# Learning to Predict Structural Vibrations

Jan van Delden[1,*], Julius Schultz[2], Christopher Blech[2], Sabine C. Langer[2], and Timo Lüddecke[1]

[1]Institute of Computer Science, University of Göttingen
[2]Institute for Acoustics and Dynamics, Technische Universität Braunschweig
[*]Correspondence: jan.vandelden@uni-goettingen.de

## Abstract

In mechanical structures like airplanes, cars and houses, noise is generated and transmitted through vibrations. To take measures to reduce this noise, vibrations need to be simulated with expensive numerical computations. Deep learning surrogate models present a promising alternative to classical numerical simulations as they can be evaluated magnitudes faster, while trading-off accuracy. To quantify such trade-offs systematically and foster the development of methods, we present a benchmark on the task of predicting the vibration of harmonically excited plates. The benchmark features a total of 12,000 plate geometries with varying forms of beadings, material, boundary conditions, load position and sizes with associated numerical solutions. To address the benchmark task, we propose a new network architecture, named Frequency-Query Operator, which predicts vibration patterns of plate geometries given a specific excitation frequency. Applying principles from operator learning and implicit models for shape encoding, our approach effectively addresses the prediction of highly variable frequency response functions occurring in dynamic systems. To quantify the prediction quality, we introduce a set of evaluation metrics and evaluate the method on our vibrating-plates benchmark. Our method outperforms Deep-ONets, Fourier Neural Operators and more traditional neural network architectures and can be used for design optimization. Code, dataset and visualizations: https://github.com/ecker-lab/Learning_Vibrating_Plates

## 1 Introduction

Humans are exposed to noise in everyday life, which is unpleasant and unhealthy in the long term [1]. Therefore, designers and engineers work on reducing noise that occurs, for example, in cars, airplanes, and houses. In this work, we specifically consider vibrations in mechanical structures as a source of sound. Vibrating structures radiate sound into the surrounding air. For example in a car, the engine causes the chassis to vibrate, which then radiates sound into the interior of the car. By reducing the vibration energy of the chassis, the noise can be reduced.

Vibrations of mechanical structures depend on the frequency of the excitation force (e.g. by the engine). A special case occurs when the excitation frequency matches an eigenfrequency of a given structure. In this case, the external force adds energy in phase with the structure's natural vibration and amplifies the motion with each cycle. This continues until the energy added equals the energy lost due to damping, resulting in large vibration amplitudes. This effect is called resonance and leads to characteristic resonance peaks in the dynamic response of the system. At resonance frequencies, due to the higher vibration amplitudes, more noise is emitted. A second distinctive feature of structural vibrations is the vibration pattern, i.e. the spatial field of vibration velocity amplitudes. With increasing frequency, these vibration patterns become more complex and exhibit more local maxima and minima (Figure 1, left) [2].

38th Conference on Neural Information Processing Systems (NeurIPS 2024).

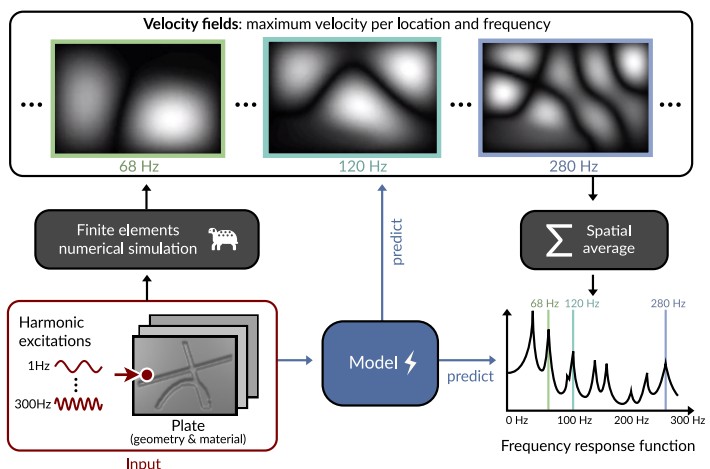 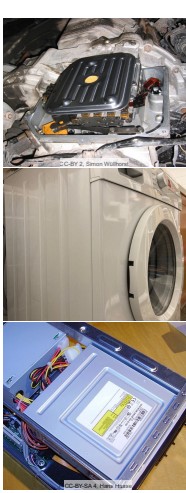

Figure 1: Left: We introduce the Vibrating Plates dataset of 12,000 samples for predicting vibration patterns based on plate geometries. A harmonic force excites the plates, causing them to vibrate. The vibration patterns of the plates are obtained through numerical simulation. Diverse architectures are evaluated on the dataset. Right: Beadings are indentations and used in many vibrating technical systems. Here, on an oil filter, a washing machine and a disk drive. They increase the structural stiffness and alter the vibration.

To reduce noise, the vibration patterns of a mechanical structure can be influenced through modifications to its design. One method is the placement of damping elements, that absorb vibrational energy and thereby reduce sound emission, but this adds weight and requires space. Another approach is introducing beadings, which are indentations in plate-like structures (Figure 1, right). Beadings increase the local stiffness of a structure, resulting in a shift in the structure's eigenfrequencies and subsequent resonance peaks. When they are well-placed, beadings can reduce the vibration energy for a range of excitation frequencies by shifting the resonance peaks out of the range. Reducing the vibration energy for a specific range of frequencies is a goal in many applications, e.g. in automotive design, where a motor excites vibrations in a range of frequencies [3].

In this work, we focus on a crucial prerequisite for targeted modifications to a design: Computing its vibrational behavior. The finite element method (FEM) is an established approach for numerically solving partial differential equations. The geometry of a design is discretized into small elements and the solution of the PDE is approximated by simple functions, e.g. polynomial functions, defined on these elements. [4, 5]. This method enables the numerical simulation of vibration patterns, but is computationally expensive. With increasing frequency and decreasing wavelength, finer meshes are required to accurately resolve the vibrations. This leads to a high increase in computational load and limits the number of designs and value of the frequencies that can be evaluated. Deep learning surrogate models could accelerate the evaluation of design candidates by several magnitudes.

Related work on predicting the solution of partial differential equations with deep learning has mostly focused on time-domain problems [e.g. 6, 7, 8]. In contrast, for our problem the change over time is not of interest. Instead, we predict steady-state vibration patterns in the frequency domain. Steady-state refers to the fact that the system vibrates harmonically and the amplitude and frequency remain constant over time since the system is in a dynamic equilibrium. Despite being practically relevant in acoustics and structural dynamics in general this problem is so far under-explored by machine learning research.

**Contributions.** To explore the potential of vibration prediction with deep learning methods, we (1) introduce a benchmark and define evaluation metrics on it, (2) evaluate a range of machine learning methods on the benchmark and (3) introduce our own method.

Our novel benchmark dataset consists of 12,000 instances of an exemplary structural mechanical system, a plate excited by a harmonic force, and their numerically computed vibrations given a range of excitation frequencies. Given a plate instance, the task is to predict the vibration patterns and frequency response. We vary material properties and the boundary conditions of the plate as well as the geometry by adding beadings. Plates with beadings are abundant in technical systems

(Figure 1, right). Plates are also often a component of more complex mechanical systems and their vibrational behavior on their own is similar to more complex systems [9, 10], making them a well-posed and scalable initial benchmark problem for deep learning methods.

To address the benchmark task, we propose a novel network architecture named Frequency-Query Operator (FQO). This model is trained to predict the resulting vibration pattern from plate geometries together with an excitation frequency query. This approach is inspired by work on operator learning for predicting the solution to partial differential equations [11] and implicit models for shape representation [e.g. 12, 13, 14], both techniques enable evaluating any point in the domain instead of a fixed grid. In our case, this enables predictions for any excitation frequency, including those not seen during training. On our vibrating-plates benchmark, the proposed FQO can accurately predict the highly variable resonances occurring in vibration patterns and outperforms DeepONet [11], Fourier Neural Operators [15] and other baselines.

## 2 Dataset and Benchmark Construction

### 2.1 Vibrating Plates Dataset

We introduce a dataset consisting of instances of aluminum plate geometries and their vibration patterns. The plates are simply supported, i.e. the edges cannot move up and down. Depending on the dataset setting, the rotational stiffness at the boundary is varied, which corresponds to free rotation or clamped edges. The plate is excited by a harmonic point force at varying positions with the excitation frequency varied between 1 and 300 Hz. While the specific setting in other mechanical engineering design tasks may differ, this setup functions as an exemplary engineering design problem. Analogous problems are the design of an air-conditioning enclosure [16], a washing machine [17] or parts of a car chassis [3]. Compared to these problems, our plate setup has two differences that allow for a comparatively easy experimental real world validation of the computed vibration patterns and do not change typical vibrational characteristics: First, exciting the plate with a point force is a common experimental setup, where a plate is excited via a shaker. Second, the condition of no rotational stiffness at the edges in comparison to clamped edges does not introduce additional uncertainty and parameters into the measurement and mirrors e.g. a bonnet of a car that rests on the chassis. Other typical types of fixation include screws or welding. In the following, we describe the specific quantity of interest of the vibration patterns, how the vibration patterns of the plate are obtained via numerical simulation and how the plate geometry and parameters are varied.

**Vibration patterns and frequency response function.** Our benchmark is designed to address a vibroacoustic engineering design problem. Therefore, the goal is to predict a quantity that best reflects the noise emitted by a mechanical structure. For a plate, a natural choice is the maximum velocity field $v_z(x, y|f)$ for a specific frequency $f$. Here, $v_z(x, y|f)$ represents the component of the velocity field orthogonal to the plate surface (in the following $\mathbf{V}(f)$ denotes the velocity field on the discrete grid). This component closely relates to how much sound is radiated, but specific details about where the velocity on the plate is highest are superfluous. Therefore, we use the mean of the squared velocity as a more compact representation and express it in a frequency response function $\mathcal{F}$, which is a function of the excitation frequency:

$$\mathcal{F}(f) = 10 \log_{10} \left( \frac{r}{A} \int_A v_z(x, y|f)^2 \, dA \right) \tag{1}$$

The square velocity is proportional to the kinetic energy and is therefore closely related to how strongly the vibration couples into a surrounding fluid and can then be perceived as airborne sound. In the above expression, $A$ is the plate area over which the velocity is averaged. The result is scaled by a reference value $r$ and converted to a decibel scale.

**Numerical simulation.** Historically, plate structures have been the subject of intense research regarding their vibrational behavior [e.g. 18, 19]. A common approach in plate modeling is to reduce the model to a two-dimensional problem with the goal to accurately describe the vibrational behavior while being computationally efficient [20]. To model the vibrational behavior of plates in this work, we use a shell formulation based on Mindlin's plate theory [18]. This theory is applicable for moderately thin plates and represents the plate using a mid-plane with constant thickness. Mindlins

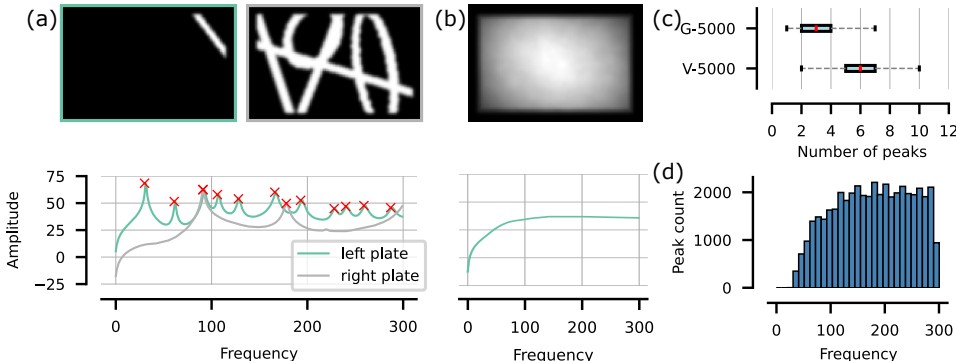

Figure 3: Dataset analysis. (a) shows two discretized plate geometries with their corresponding frequency response, the red crosses mark the detected peaks. (b) shows the mean plate design and frequency response. (c) shows number of peaks in different dataset settings. (d) shows the distribution of the peaks over the frequencies.

plate theory is a standard choice in many engineering applications and has been experimentally validated [21, 22].

We apply the finite element method to solve the shell formulation and simulate the vibrational behavior of the plate [4] (Figure 2). This involves partitioning the plate geometry into discrete elements and approximating the solution on these elements by simple ansatzfunctions. By choosing a sufficiently large number of elements, the solution converges to the exact solution of the model [23]. We discretize the plate with a regular grid and use triangular elements in the domain to allow a flexible representation of beadings. The discretization is sufficient to resolve wave lengths in the plate structure, but limits the detail that can be represented with the beading patterns. After discretizing the plate, the PDE is integrated over the elements and a linear system of equations is derived. This linear system describes the dynamics of the discretized structure and is solved with a direct solver. We perform the computations with a specialized FEM software for acoustics [24]. Further details on the setup and mechanical model are given in Appendix A.1.

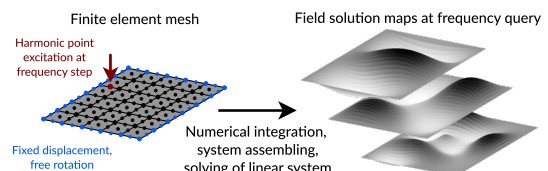

Figure 2: Process of the finite element solution in frequency domain in order to compute the velocity field at each frequency query.

**Dataset variations.** The plate instances are varied in two settings: For the V-5000 setting, we generate random beading patterns consisting of 1 - 3 lines and 0 - 2 ellipses. Also, the width of the beading-elements is randomly varied. The size of the plates as well as material, boundary and loading parameters are fixed. For the G-5000 setting, we apply the same beading pattern variation and additionally vary the plate geometry (length, width and thickness) as well as the damping loss factor, rotational stiffness at the boundary and forcing position. For each setting, 5000 instances for training and validation are generated. 1000 further instances are generated as a test set and are not used during training or to select a model. Further details are given in Appendix A.2.

**Dataset analysis.** The mean plate design shows a close to uniform distribution, with a margin at the plate's edge (see Figure 3b). With a greater proportion of beaded area in a given plate, the number of peaks tends to decrease (see Figure 3a). This is due to additional beadings stiffening the plates, and it represents an interesting trait specific to our problem. The density of peaks is related to the frequency. As the frequency increases, so does the peak density. Starting from around 120 Hz the peak density plateaus (see Figure 3d). The average number of peaks in the G-5000 setting is smaller than in the V-5000 setting. This is influenced by the on average smaller plates being stiffer and therefore having less peaks in the frequency range (see Figure 3c).

## 2.2 Evaluation

Before computing our metrics, we perform the following preprocessing steps to address numerical issues as well as facilitate an easier interpretation of the evaluation metrics. We normalize the fre-

quency response and the velocity fields. To do this, we first take the log of the velocity fields, to align it with the dB-scale of the frequency response. Then, we subtract the mean per frequency over all samples (depicted in Figure 3b for frequency response) and then divide by the overall standard deviation across all frequencies and samples. Small changes in the beading pattern can cause frequency shifts, potentially pushing peaks out of the considered frequency band. To reduce the effect of such edge cases, we predict frequency responses between 1 and 300 Hz but evaluate on the frequency band between 1 and 250 Hz.

We propose three complementary metrics to measure the quality of the frequency response predictions.

**Mean squared error.** The *mean squared error (MSE)* is a well-known regression error measure: For the global deviation we compare the predicted $\hat{\mathcal{F}}(f)$ and numerically computed frequency response $\mathcal{F}(f)$ by the MSE error $\mathcal{E}_{\mathrm{MSE}} = \sum_i (\hat{\mathcal{F}}(f_i) - \mathcal{F}(f_i))^2$.

**Earth mover distance.** The *earth mover distance* [25, 26] expresses the work needed to transmute a distribution $P$ into another distribution $Q$. As a first step, the optimal flow $\hat{\gamma}$ is identified. Based on $\hat{\gamma}$ the earth mover distance is expressed as follows:

$$\mathcal{E}_{\mathrm{EMD}}(P, Q) = \frac{\sum_{i,j} \hat{\gamma}_{ij} \cdot d_{ij}}{\sum_{i,j} \hat{\gamma}_{ij}} \quad \text{with } \hat{\gamma} = \min_{\gamma} \sum_{i,j} \gamma_{ij} \cdot d_{ij}$$

where $d_{ij}$ is the distance between bins $i$ and $j$ in $P$ and $Q$. Correspondingly, $\gamma_{ij}$ is the flow between bins i and j. We calculate the $\mathcal{E}_{\mathrm{EMD}}$ based on the original amplitudes in $m/s$ that have not been transformed to the log-scale (dB) and normalize these amplitudes with the sum over all frequencies. As a consequence and unlike the MSE, $\mathcal{E}_{\mathrm{EMD}}$ is invariant to the mean amplitude and only considers the shape of the frequency response. In this form, our metric is equivalent to the $W_1$ Wasserstein metric [27, 28].

**Peak frequency error.** To specifically address the prediction of resonance peaks, which are particularly relevant for noise emission, we introduce a third metric called *peak frequency error*. The metric answers two questions: (1) Does the predicted frequency response contain the same number of resonance peaks as the true response? (2) How far are corresponding ground truth and prediction peaks shifted against each other? To this end, we set up an algorithm that starts by detecting a set of peaks $K$ in the ground truth and a set of peaks $\hat{K}$ in the prediction using the `find_peaks` function in scipy [29] (examples in Appendix B). Then, we match these peaks pairwise using the Hungarian algorithm [30] based on the distance between the frequencies of the peaks $\mathcal{E}_{\mathrm{F}}$. This allows us to determine the ratio between predicted and actual peaks $\frac{|\hat{K}|}{|K|}$ and $\frac{|K|}{|\hat{K}|}$. To equally penalize predicting too many and too few peaks we consider the minimum of both ratios: $\mathcal{E}_{\mathrm{PEAKS}} = 1 - \min\{\frac{|\hat{K}|}{|K|}, \frac{|K|}{|\hat{K}|}\}$.

## 3 Predicting Vibrations with Neural Networks

We propose a method to predict the frequency response, $\mathcal{F}_{\mathbf{g},\mathbf{m}}(f)$, for plates characterized by their geometry $\mathbf{g}$ (influenced by beading patterns) and scalar parameters $\mathbf{m}$ (height, width, thickness, damping loss factor, rotational stiffness at boundary, loading position). This process involves two steps: (1) First, the input $\mathbf{g}$ and $\mathbf{m}$ are encoded by an encoder $\Phi$. Because $\mathbf{g}$ is defined on a regular grid, standard image processing architectures are suitable. (2) Frequency response predictions are generated for specific excitation frequencies $f$ by a decoder $\Psi$ (Figure 4). The computation can then be expressed as:

$$\Psi(\Phi(\mathbf{g}, \mathbf{m}), f) = \hat{\mathcal{F}}_{\mathbf{g},\mathbf{m}}(f) \tag{2}$$

This problem formulation, training a neural network to predict a function and evaluating this function, given some input values, is a common paradigm in operator learning [11]. It allows for the evaluation of any frequency query $f$, even if it has not been part of the training data. In contrast, predicting frequencies on a fixed grid only allows for the evaluation of those frequencies. This formulation shares similarities with implicit models, for instance by [13] in the context of 3d shape prediction. Based on this, we investigate the following central aspects of our architecture:

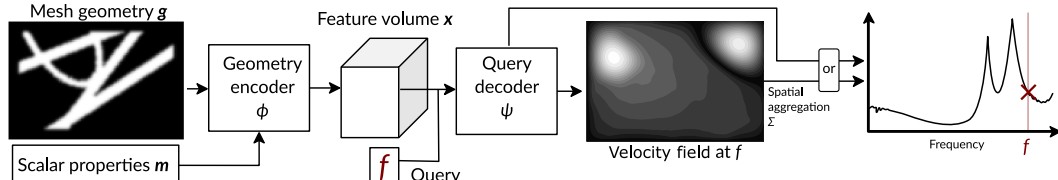

Figure 4: Frequency-Query Operator method. The geometry encoder takes the mesh geometry and the scalar properties as input. The resulting feature volume along with a frequency query is passed to the query decoder, that either predicts a velocity field or directly a frequency response. The velocity field is aggregated to arrive at the frequency response at the query frequency $f$.

**Q1 - Frequency-query approach:**  Vibrations are dominated by resonance peaks at specific frequencies. The resonance frequencies vary strongly across instances. An implicit or operator learning approach has been shown to be able to deal with high variation better in other contexts. In the context of vibration prediction, a frequency-query approach could be employed to generate predictions for one specific frequency.

**Q2 - ViT encoder:**  Image processing architectures based on convolutions encode local features. In contrast, vision transformers have a global receptive field size from early layers. As vibrations are determined by the full geometry, we expect vision transformers to perform better.

**Q3 - Velocity field prediction:**  We can train networks to either directly predict the aggregate frequency response $\mathcal{F}$ or to predict the velocity field $\mathbf{V}$ and compute $\mathcal{F}$ from $\mathbf{V}$ via Equation 1. For predicting the velocity field, much richer training data is available, since it describes a field over the plate instead of the scalar frequency response. Most of this information is not represented in the frequency response.

In the following, we describe architectural variations explored for these aspects.

## 3.1 Geometry Encoder $\Phi$

To parse the plate geometry into a feature vector, we employ three variants: ResNet18 [31, RN18], a vision transformer [32, 33, ViT] and the encoder part of a UNet [34]. For the RN18, we replace batch normalization with layer normalization [35], as we found this to work substantially better. Compared to the CNN-based RN18, the ViT architecture supports interactions across different image regions in early layers. For both, the RN18 and the ViT encoder, we obtain a feature vector $\mathbf{x}$ by average pooling the last feature map. Since the UNet generates velocity fields, no pooling is applied.

**FiLM conditioning.**  For including the scalar parameters $\mathbf{m}$, we introduce a film layer [36]. The film layer first encodes the scalar parameters with a linear layer. The resulting encoding is then multiplied element-wise with the feature of the encoder and a bias is added. This operation is applied before the last layer of the geometry encoder (UNet) or after it (RN18, ViT).

## 3.2 Decoder $\Psi$

**FQO-RN18 and FQO-ViT: Predicting $\mathcal{F}(f)$ directly.**  Having obtained an encoding of the plate geometry and properties $\mathbf{x}$, a decoder now takes this as well as a frequency query as input and maps them towards a prediction. For the RN18 and ViT geometry encoders, the decoder is implemented by an MLP taking both $\mathbf{x}$ and a scalar frequency value $f$ as input to predict the response for that specific query frequency, i.e. $\Psi(\mathbf{x}, f) \in \mathbb{R}$. The frequency query is merged to $\mathbf{x}$ by a film layer [36]. By querying the decoder with all frequencies individually, we obtain results for the frequency band between 1 and 300 Hz. The MLP has six hidden layers with 512 dimensions each and ReLU activations.

**FQO-UNet: Predicting $\mathcal{F}(f)$ through the velocity field $\mathbf{V}(f)$.**  To incorporate physics-based contraints and take advantage of the larger amount of available data, we employ a UNet to predict the velocity fields, $\mathbf{V}(f)$. From $\mathbf{V}(f)$, we derive the frequency response $\mathcal{F}(f)$ (analogous to Equation 1). A frequency query, introduced via a FiLM layer after the encoder, enables frequency-specific predictions. To reduce the memory and computation demands per geometry during training, we select a random subset of $k$ frequency queries per geometry in a batch, with $k < 300$. If not otherwise specified, $k$ is set to 50.

Table 1: Test results for frequency response prediction. Column **VF** indicates if $\mathcal{F}$ is indirectly predicted through the velocity field (Q3), column **FQ** indicates if frequency queries (Q1) are used. Q1 to Q3 refer to the model components described in Section 3.

| | **FQ** | **VF** | **V-5000** | | | | **G-5000** | | | |
|---|---|---|---|---|---|---|---|---|---|---|
| | | | $\mathcal{E}_{\mathrm{MSE}}$ | $\mathcal{E}_{\mathrm{EMD}}$ | $\mathcal{E}_{\mathrm{PEAKS}}$ | $\mathcal{E}_{\mathrm{F}}$ | $\mathcal{E}_{\mathrm{MSE}}$ | $\mathcal{E}_{\mathrm{EMD}}$ | $\mathcal{E}_{\mathrm{PEAKS}}$ | $\mathcal{E}_{\mathrm{F}}$ |
| Baselines | | | | | | | | | | |
| $k$-NN | - | - | 0.63 | 21.50 | 0.45 | 8.7 | 0.88 | 32.48 | 0.68 | 21.0 |
| RN18 + FNO | - | - | 0.42 | 10.76 | 0.34 | 5.6 | 0.28 | 14.12 | 0.21 | 6.1 |
| DeepONet | ✓ | - | 0.49 | 16.91 | 0.48 | 5.4 | 0.44 | 23.05 | 0.57 | 9.9 |
| FNO (velocity field) | - | ✓ | 0.47 | 13.10 | 0.36 | 6.3 | 0.49 | 21.16 | 0.39 | 10.7 |
| Grid-RN18 | - | - | 0.44 | 13.29 | 0.36 | 5.4 | 0.30 | 14.95 | 0.26 | 6.5 |
| FQO-RN18 (Q1) | ✓ | - | 0.32 | 10.70 | 0.17 | 5.3 | 0.24 | 13.51 | 0.13 | 5.1 |
| FQO-ViT (Q2) | ✓ | - | 0.68 | 20.96 | 0.54 | 7.1 | 0.52 | 24.34 | 0.49 | 11.5 |
| Grid-UNet | - | ✓ | 0.19 | 7.57 | 0.24 | 2.7 | 0.17 | 9.41 | 0.14 | 4.6 |
| **FQO-UNet** | ✓ | ✓ | 0.08 | 4.24 | 0.07 | 1.7 | 0.11 | 7.47 | 0.08 | 3.1 |

**Grid-Unet and Grid-RN18: Predicting $\mathcal{F}$ for a fixed grid of frequencies.** To ablate the frequency-query approach, we employ two variations of the FQO-RN18 and FQO-UNet architectures, that do not employ frequency queries. They instead generate predictions for 1-300 Hz at once. This is done by setting the output size of the respective last layer to 300.

### 3.3 Baseline Methods

We further report baseline results on the following alternative methods: A $k$-Nearest Neighbors regressor, that finds the nearest neighbors in the latent space of an autoencoder. DeepONet [11], with a RN18 as backbone and a MLP to encode the query frequencies as a branch net. Two architectures based on Fourier Neural Operators [15]. One employing an FNO as a replacement for the query-based decoder based on RN18 features. The second directly takes the input geometry and is trained to map it to the velocity fields.

### 3.4 Training

All methods are trained in a data-driven fashion for 500 epochs on the training dataset of 5000 samples. 500 samples from the training dataset are excluded and employed for validation. We report evaluation results on the previously unseen test set consisting of 1000 additional samples.
For methods that predict $\mathbf{V}(f)$, i.e. UNet based methods and the FNO variation, the training loss is set to $L_{\mathbf{V}}$ where $L_{\mathbf{V}}$ represents the MSE on the log-transformed, normalized squared velocity field (Ablation on loss function in Appendix D). For methods that directly predict $\mathcal{F}$, the loss is set to $L_F$, the MSE on the normalized frequency response. Choosing the log-transformed quantities enables the loss to be sensitive to errors outside of resonance frequencies. Otherwise, such errors would have little influence on the total loss, as their magnitude is much lower. See Appendix C for further details on the architectures and training procedure.

## 4 Experiments

We train the architecture variations and baseline methods on the Vibrating Plates dataset (see Table 1). To assess which architecture aspects described in Section 3 are beneficial, we perform the following comparisons. Regarding Q1 (frequency-query approach), the Frequency-Query Operator variations consistently yield better predictions than equivalent grid-based methods, where responses for all frequencies are predicted at once: The $\mathcal{E}_{\mathrm{MSE}}$ and the $\mathcal{E}_{\mathrm{EMD}}$ are lower, more peaks are reproduced, and the peak positions are more precise. Regarding Q3 (velocity field prediction), predicting the velocity fields and then transforming them to the frequency response leads to better results than directly predicting the frequency response. Specifically, the UNet based architectures strongly outperform all alternatives, which we attribute to the richer training data of velocity fields. Regarding Q2, the ViT encoder leads to worse results than the CNN-based encoders.
All evaluated baseline methods achieve comparatively worse results than our proposed methods. Despite using the same RN18 geometry encoder as FQO-RN18, DeepONet [11] performs worse. We assume that this is due to incorporating frequency information through a single weighted summation, which limits the model's expressivity [37]. In contrast, FQO-RN18 introduces the queried frequency

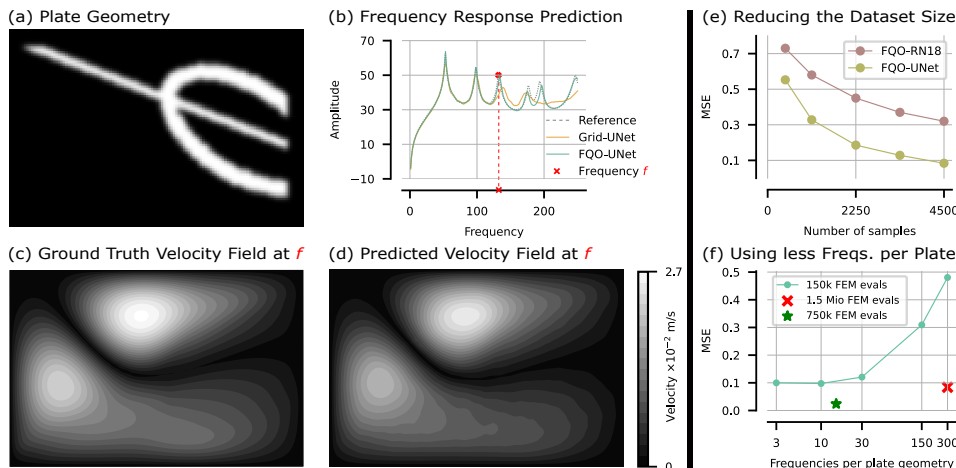

Figure 5: Results. (b) to (d) show the velocity field at one frequency and prediction for the plate geometry in (a) from FQO-UNet. (e) shows the test MSE for training two methods with reduced numbers of samples from V-5000. (f) shows effects of different data generation strategies. The blue line is an isoconture for a fixed compute budget of 150,000 data points, with varying number of frequencies per plate geometry. The green star represents using a larger dataset at 15 frequencies per plate (half of V-5000). The red cross represents a model trained on V-5000. Training with fewer frequencies per plate is more efficient.

earlier into the model. Two Fourier Neural Operator [15] baseline methods are evaluated: the first, RN18 + FNO, which substitutes the query-based decoder with an FNO decoder, underperforms compared to FQO-RN18 on both datasets. The second FNO baseline, trained directly to predict velocity fields, yields poorer results.

Results for the G-5000 setting are slightly worse than for the V-5000 setting. The difference is surprising small considering the seven additional varied parameters in the G-5000 setting. One reason might be the average number of peaks in the frequency response: the plates in G-5000 are on average smaller and because of this stiffer, leading to fewer peaks (on average 3.9 in G-5000 and 5.9 in V-5000). This interpretation is supported by the fact that the average error becomes higher with increasing frequency and thus increasing peak density (Figure 3d).

Looking at a prediction example (Figure 5a-d) for our best model, FQO-UNet, the predicted velocity field has subtle differences to the ground truth. The prediction captures the two modes and their shape quite well, but the shape is slightly less regular than in the reference. Despite that, the resulting frequency response prediction at $f = 131$ is close to the FEM reference. In comparison to the grid-based prediction, where peaks tend to be blurry, the frequency response peaks generated by FQO-UNet are more pronounced. Additional visualizations are provided in Appendix E.3 and in the code repository. For the best architecture in our experiments, FQO-UNet, we report mean and standard deviation results for multiple runs in Appendix E.2 and provide an ablation of model size for the FQO-UNet and Grid-UNet architectures in Appendix D.

**Transfer learning.** To quantify to which degree features learned on a subset of the design space transfer to a different subset, the V-5000 setting is split into two equally-sized parts based on the number of mesh elements that are part of a beading. The "more beadings" set contains only 5.1 peaks on average because the plates are stiffened by the beadings, compared to 6.7 peaks on average for the "less beadings" set. The training on plates with less beadings leads to a smaller drop in prediction quality (see Table 2). This indicates that training on data with more complex frequency responses might be more efficient. In addition, we train a single model on both G-5000 and V-5000. Performance increases, indicating that training can benefit from training with data based on similar mechanical models (Table 3).

**Sample efficiency.** We train the FQO-UNet and the FQO-RN18 with reduced numbers of samples (see Figure 5e). It is notable, that the FQO-UNet with a quarter of the training data achieves nearly the same prediction quality as the FQO-RN18 with full training data. This highlights the benefit of including the velocity fields into the training process. Quantitative results are given in Appendix E.1 for both dataset settings.

Table 2: Transfer learning performance: We split V-5000 into two halves based on amount of beadings and evaluate transfer learning performance across these splits: training subset $\mapsto$ test subset. The gray rows denote test results on the original subset that has been used for training.

| | less beadings $\mapsto$ more beadings | | | | more beadings $\mapsto$ less beadings | | | |
| --- | --- | --- | --- | --- | --- | --- | --- | --- |
| | $\mathcal{E}_{\mathrm{MSE}}$ | $\mathcal{E}_{\mathrm{EMD}}$ | $\mathcal{E}_{\mathrm{PEAKS}}$ | $\mathcal{E}_{\mathrm{F}}$ | $\mathcal{E}_{\mathrm{MSE}}$ | $\mathcal{E}_{\mathrm{EMD}}$ | $\mathcal{E}_{\mathrm{PEAKS}}$ | $\mathcal{E}_{\mathrm{F}}$ |
| FQO-RN18 | 0.61 | 16.19 | 0.20 | 9.3 | 0.82 | 15.79 | 0.36 | 8.3 |
| (origin) | 0.33 | 10.48 | 0.18 | 5.0 | 0.42 | 12.00 | 0.29 | 5.8 |
| FQO-UNet | 0.39 | 11.17 | 0.21 | 5.6 | 0.54 | 12.02 | 0.25 | 5.5 |
| (origin) | 0.18 | 8.68 | 0.19 | 2.6 | 0.17 | 7.83 | 0.13 | 3.0 |

Table 3: A FQO-UNet is trained in parallel on batches from V-5000 and G-5000 and evaluated on the G-5000 test set. Performance increases in all metrics.

| | $\mathcal{E}_{\mathrm{MSE}}$ | $\mathcal{E}_{\mathrm{EMD}}$ | $\mathcal{E}_{\mathrm{PEAKS}}$ | $\mathcal{E}_{\mathrm{F}}$ |
| --- | --- | --- | --- | --- |
| G-5000 | 0.111 | 7.47 | 0.079 | 3.1 |
| G-5000 + V-5000 | 0.093 | 6.97 | 0.071 | 2.9 |

We further investigate the optimal ratio of numbers of frequencies per geometry and total number of geometries, by generating an additional dataset in the V-5000 setting consisting of 50,000 plate geometries but with only 15 frequency evaluations per geometry. These frequencies are uniformly spaced with a random starting frequency. Reducing the frequencies per geometry drastically increases the data efficiency of our method. With a tenth of data points compared to our original dataset, the MSE metric approaches the original value (Figure 5f, quantitative results in Appendix E.1).

**Design optimization.** We investigate the potential of our FQO-UNet to be used for optimizing a beading pattern for reduced vibrations in a specified frequency range. Following the approach described in [38], to generate plates with reduced vibrations, a diffusion model trained to generate novel beading patterns is combined with gradient information from our FQO-UNet as follows: A gradient on the pixels of the input beading pattern is obtained by passing a beading pattern through the network, computing the sum of the predicted frequency response as a loss and then performing backpropagation to the input beading pattern. This gradient is then used to guide the diffusion model to generate beading patterns with reduced vibrations. We optimize beading patterns to reduce vibrations between 100 and 200 Hz using the FQO-UNet trained on the V-5000 dataset (Figure 6). Resulting plates have a lower mean frequency response in the targeted range than any plate in the training dataset.

## 5 Related Work

**Acoustics.** While research on surrogate models for the spatio-temporal evolution of vector fields is fairly common [39, 40, 41], directly predicting frequency responses through neural networks is an understudied problem. A general CNN architecture is applied in [42] to calibrate the parameters of an analytical model for a composite column on a shake table. The data includes spectrograms representing the structural response in time-frequency domain. The frequency-domain response of acoustic metamaterials is considered in a material design task by conditional generative adversarial networks or reinforcement learning [43, 44, 45]. The frequency response of a multi-mass oscillator is

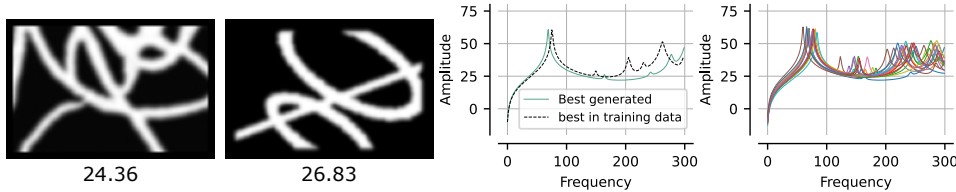

Figure 6: Design optimization. Exemplary generation result with lowest mean response between 100 Hz and 200 Hz out of 32 generations (left, mean response below). Plate with lowest response out of all 5000 training examples from V-5000 (middle left). Comparison of responses from left plates (middle right). Responses from 16 generated plates (right).

predicted with transformer-based methods [46]. Within the context of aeroacoustics, the propagation of a two-dimensional acoustic wave while considering sound-scattering obstacles is predicted in time-domain by a CNN [47, 48]. A review of machine learning in acoustics is given by [49]. Several acoustic benchmarks for numerical methods are available [50], however, these benchmarks do not systematically vary input geometries, making them not directly applicable to data-driven models.

**Scientific machine learning.**   Data-driven machine learning techniques were successfully applied in many different disciplines within engineering and applied science; for example for alloy discovery [51], crystal structure prediction [52], climate modeling [53] and protein folding [54]. A popular use case for data-driven methods is to accelerate fluid dynamics, governed by the Navier-Stokes equations [39, 40, 55, 56, 57].

The question of how to structure and train neural networks for predicting the solution of partial differential equations (PDE) has been the topic of intense research. Many methods investigate the inclusion of physics informed loss terms [58, 59, 60, 56, 61]. Some methods directly solve PDEs with neural networks as a surrogate model [62, 63]. Graph neural networks are often employed, e.g. for interaction of rigid and deformable objects as well as fluids [64, 65].

**Operator learning and implicit models.**   A promising avenue of research for incorporating inductive biases for physical models has been operator learning [11, 15, 66, 37, 41]. Operator learning structures neural networks such that they implement a function that can be evaluated at real values instead of a fixed discrete grid. DeepONet [11] implements operator learning by taking the value at which it is evaluated as an input and processes this value in a separate branch. Fourier Neural Operators [15] use a point-wise mapping to a latent space which is processed through a sequence of individual layers in Fourier space before being projected to the output space.

Implicit models (or coordinate-based representation) are models where location is utilized as an input to obtain a location-specific prediction, instead of predicting the entire grid at once and thus fit in the operator learning paradigm. Such models were used to represent shapes [12, 67, 68, 13], later their representations were improved [69, 70] and adapted for representing neural radiance fields (NeRFs) [71, 14]. Our method applies techniques from these implicit models to operator learning.

# 6   Conclusion

We introduced the problem of predicting structural vibrations and associated frequency response functions of mechanical systems. Unlike other benchmarks for deep learning surrogate models, this task necessitates predicting a steady-state solution that remains constant over time, but varies across different excitation frequencies. To this end, we created the Vibrating Plates dataset and benchmark and provide reference scores for several methods. Our Frequency-Query Operator method addresses the benchmark and achieves better results than the DeepONet and FNO baselines. We find that query-based approaches and the indirect prediction of a mean frequency response through predicted field quantities lead to better results. Surrogate models as shown in this work can greatly accelerate the prediction of physical quantities over the finite element method: Our models achieved a speed-up of around 4 to 6 orders of magnitude (see Appendix C), which makes tasks such as design optimization feasible. This efficiency, however, depends on the availability of enough pre-generated training data and requires model training. We further investigated effects of changing the composition of the training dataset and found that using less frequencies per plate and more different plates positively impacts prediction accuracy.

**Limitations and future work.**   Our dataset and method serve as an initial step in the development of surrogate models for vibration prediction. The dataset focuses on plates, a common geometric primitive used in a great number of applications. However, many structures beyond plates exist, involving curved shells, multi-component geometries and complex material parameters. While some results from our study might transfer to these cases, more flexible architectures, able to deal with 3D data, would be needed. Different mechanical models, might also produce more complex frequency responses with e.g. more closely spaced modes, making the prediction task more challenging. As more complex geometries incur higher computational costs of FEM simulations, key questions are how to enhance sample-efficiency further, for example through transfer learning. A further limitation is the manufacturability of the considered beading patterns. The plate beadings could in principle be manufactured by deep drawing of sheet metal, but would require specifically designed stamps.

**Acknowledgements.**   This research is funded by the Deutsche Forschungsgemeinschaft (DFG, German Research Foundation), project number 501927736, within the DFG Priority Programme

2353: Daring More Intelligence - Design Assistants in Mechanics and Dynamics'. The authors gratefully acknowledge the computing time made available to them on the high-performance computers HLRN-IV at GWDG at the NHR Centers NHR@Göttingen. These centers are jointly supported by the German Federal Ministry of Education and Research and the German state governments participating in the NHR (www.nhr-verein.de/unsere-partner).

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

# A   Dataset Construction

## A.1   The Mechanical Model

In the following, we give a technical description of the mechanical model that is applied to generate the datasets. For moderately thin plates, the plate theory by Mindlin is a valid differential equation [18]:

$$B \, \nabla^4 u_z - \omega^2 \rho_s h \, u_z + \omega^2 \left( \frac{B\rho_s}{G} + \rho_s I \right) \, \nabla^2 u_z + \omega^4 I \frac{\rho_s^2}{G} \, u_z = p_l$$

This equation is combined with a disc formulation for in-plane loads in order to receive a shell formulation for the mechanical description of arbitrarily formed moderately thin structures considering in-plane and transverse loads. The plate part is the dominating and important part for resolving bending waves. In the equation, $u_z$ denotes the normal displacement of the plate structure as degree of freedom of interest. $B$ represents the bending stiffness, $\rho_s$ the density, $h$ the thickness, $G$ the shear modulus and $I$ the moment of inertia. The angular frequency $\omega$ is defined as $\omega = 2\pi f$. The right hand-side excitation $p_l$ describes an applied pressure load, which is converted to point forces through integration. As boundary conditions we apply homogeneous dirichlet boundary conditions, i.e. $u_z(x) = 0$ on the boundary and include a rotational stiffness at the boundary to model different boundary conditions, ranging from free rotating to clamped plates. The equation is transformed into a weak integral formulation by weighted residuals, discretized using finite elements and integrated numerically. In particular, we use triangular shell elements with 3 nodes and linear ansatz functions. The integration delivers the sparse linear system of equations. This linear system is solved using the direct solver MUMPS [72] with a specialized FEM implementation for acoustics [24]. The discretization is chosen, such that the bending waves are resolved with a minimum of 10 nodes. The bending wave length $\lambda_B$ of a plate can be calculated by

$$\lambda_B = \sqrt{\frac{2\pi}{f}} \sqrt[4]{\frac{Et^2}{12(1 - \nu^2)\rho}},$$

where $E$ is the Young's modulus, $t$ the thickness, $\nu$ the Poisson ratio and $\rho$ the density of the plate. The final discretization is set to $181 \times 121$ for G-5000 and $121 \times 81$ for V-5000, which is sufficient for convergence.

## A.2   Datasets

The exact physical setting and variation of our mechanical model to form the V-5000 and G-5000 datasets are given in Table 4, Table 6 and Table 5. Both dataset settings contain 6000 samples, each consisting of a plate geometry with associated physical and material parameters and the computed velocity fields $\mathbf{V}(f)$ and frequency response $\mathcal{F}(f)$ for frequencies $f$ 1 to 300 Hz. 1000 samples from the 6000 samples are selected as a test dataset and not considered during neural network training or validation. Computing a single sample out of the 6000 samples takes 2 minutes and 19 seconds on a machine with a 2 Ghz CPU (20 physical cores).

Table 4: Dataset settings. Width is the width of lines and ellipses in mm. Properties. (prop.) involves plate size, thickness, material, boundary and loading properties.

|         |       | Sample space |          |         | Sample number |      |
|---------|-------|--------------|----------|---------|---------------|------|
| Setting | Prop. | Lines        | Ellipses | Width   | Train         | Test |
| V-5000  | fix   | 1 - 3        | 0 - 2    | 30 - 70 | 5000          | 1000 |
| G-5000  | vary  | 1 - 3        | 0 - 2    | 40 - 60 | 5000          | 1000 |

For the G-5000 dataset, the geometry, material, boundary condition and loading parameters are varied. The effect of two of the material parameters is visualized in Figure 7. Increasing the damping reduces amplitudes at resonance peaks but does not shift the overall form of the frequency response. Increasing the thickness increases the overall stiffness and shifts resonance peaks towards higher frequencies in a less regular manner. For boundary condition variation, we include one rotational stiffness parameter, which models the rotational stiffness along the x-axis at the lower and upper edge and along the y-axis at the left and right edge. The rotational stiffness is added at the respective rotational degree of freedom at the boundaries and varied as given in Table 6.

Table 5: Geometry and material parameters for V-5000 and G-5000 datasets.

| | Geometry | | | Material (Aluminum) | | | |
|---|---|---|---|---|---|---|---|
| | length | width | thickness | density | Young's mod. | Poisson ratio | loss factor |
| V-5000 | 0.9 m | 0.6 m | 0.003 m | 2700 kg/m$^3$ | 7e10 N/m$^2$ | 0.3 | 0.02 |
| G-5000 | 0.6 - 0.9 m | 0.4 - 0.6 m | 0.002 - 0.004 m | 2700 kg/m$^3$ | 7e10 N/m$^2$ | 0.3 | 0.01 - 0.03 |

Table 6: Loading and boundary condition parameters for V-5000 and G-5000 datasets.

| | Loading (Point force) | | Boundary condition (rot. stiffness) |
|---|---|---|---|
| | x-position | y-position | $c_{ry}/c_{rx}$ |
| V-5000 | 0.36 m | 0.225 m | 0.0 Nm |
| G-5000 | 0.18 - 0.72 m | 0.12 - 0.48 m | 0.0 - 100 Nm |

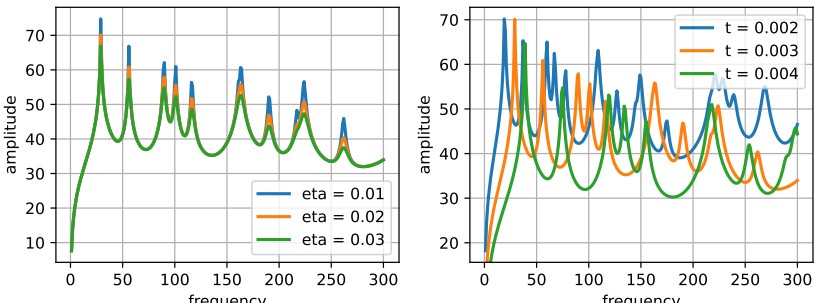

Figure 7: One-at-a-time parameter variation of the thickness parameter and the damping loss factor. Increasing the damping reduces the amplitudes at the resonance peaks. Increasing the plate thickness increases the stiffness of the plate and thus shifts the resonance peaks towards higher frequencies

# B Metrics - Peak Frequency Error

We provide examples of the find_peak operation which serves as the basis for the peak frequency error on ground truth (Fig. 8) and predictions (Fig. 9, using a kNN baseline) and visualize the matched peaks for calculating the peak frequency error (Fig. 10). Note that find_peaks is run with the prominence threshold set to 0.5 meaning that the peak must be at least 0.5 units higher than their surroundings.

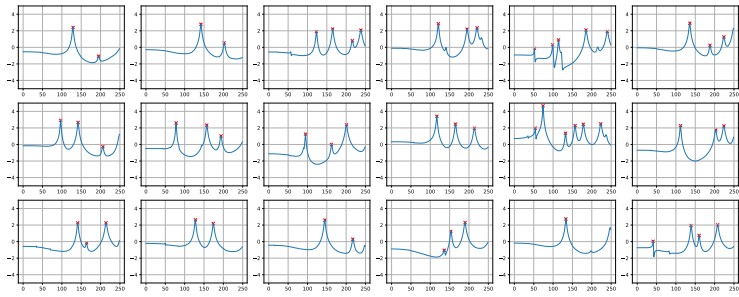

Figure 8: Find peak results on random ground truth samples.

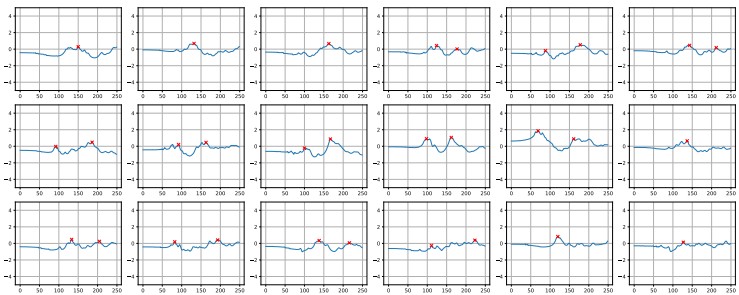

Figure 9: Find peak results on predictions.

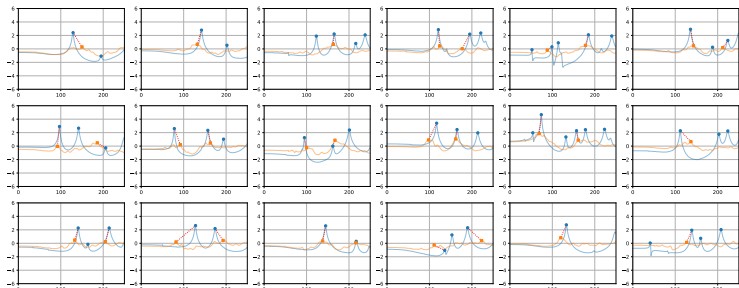

Figure 10: Visualization of the matching between ground truth (blue) and prediction (orange) peaks. Matched peaks are indicated in red.

# C   Architectures

In the following, our neural network architectures and the training procedure are described. The training and neural networks can be reproduced based on the publicly available code repository. To give an overall impression of the employed models, Table 7 gives an overview of the number of parameters and the speed for a forward pass prediction.

Table 7: Model size and speed comparison for a forward pass of a batch of 16 plate geometries on an A100 GPU. In comparison, solving one geometry via FEM takes 2 minutes and 19 seconds. The slowest deep learning method is then around 6000 times faster.

|                      | # weights in Mio | Time (s)        |
|----------------------|------------------|-----------------|
| RN18 + FNO           | 11.8             | 0.014           |
| DeepONet             | 11.3             | 0.005           |
| FNO (velocity field) | 134              | 0.008           |
| Grid-RN18            | 12.9             | 0.005           |
| FQO-RN18             | 12.7             | 0.005           |
| FQO-ViT              | 9.28             | 0.013           |
| Grid-UNet            | 27.9             | 0.010           |
| FQO-UNet             | 7.1              | 0.338           |
| FEM (20 CPU cores)   | -                | $\sim 2224.000$ |

## C.1   Frequency-Query-based Methods

**Predicting $\mathcal{F}(f)$ directly: FQO-RN18 and FQO-ViT.**   To directly predict the frequency response instead of predicting the velocity fields and then transforming it to the frequency response, we use a ResNet [31] and a vision transformer (ViT) [32] as geometry encoders. For the ResNet, we opt for the ResNet18 backbone. We replace batch normalization with layer normalization [35], as we found this to work substantially better. In addition, we employ the vision transformer (ViT) architecture [32]. The ViT supports interactions across different image regions in early layers. We use a variation of the ViT-Base configuration with a reduced token size of 192, an intermediate size of 768 and three attention heads. For both the RN18 and the ViT encoder, we obtain the $d$-dimensional global feature $\mathbf{x}$ through average pooling from the last feature map or the encoded tokens. Scalar parameters are introduced to the encoding by a film layer. As a decoder, we employ an MLP. The MLP $r$ takes both $\mathbf{x}$ and a scalar frequency value $f$ as input to predict the response for that specific query frequency, i.e. $r(\mathbf{x}, f) \in \mathbb{R}$. The frequency query is introduced by a film layer to $\mathbf{x}$. By querying the decoder with all frequencies individually, we obtain results for the frequency band between 1 and 300 Hz. The MLP has six hidden layers with 512 dimensions each and ReLU activation functions.

**Predicting $\mathcal{F}(f)$ through the velocity field $\mathbf{V}(f)$: FQO-UNet.**   To predict $\mathbf{V}(f)$ instead of directly predicting $\mathcal{F}(f)$, we employ a UNet. The plate geometry is encoded by the UNet encoder and the scalar material parameters are introduced before the last contraction block of the UNet. A frequency query is introduced after the encoder again by a film layer and the decoder then produces predictions of size $40 \times 60$, which is sufficient to capture the structure and modes of the velocity fields. Since the decoder has to be evaluated for each frequency query individually, we opt to map to predictions for five velocity fields per query.
The UNet consists of three contraction blocks, two spatial-shape-preserving blocks and two expansion blocks. Additionally, two self-attention layers are included in the encoder and one self attention layer in the decoder. To ensure global features are included in the full feature volume after the encoder, adaptive average pooling is applied to the feature volume and the result is concatenated to the feature volume.

## C.2   Grid-based Methods

To provide a direct comparison to the query-based approach, two methods that predict all frequency responses at once are tested.

**Grid-RN18.** The same RN18 is used to generate a global feature $\mathbf{x}$ as in the FQO-RN18. Given $\mathbf{x}$, an MLP $r$ predicts the frequency response on a fixed 1 Hz interval grid, with $r(\mathbf{x}) \in \mathbb{R}^{300}$. We employ six hidden layers with 512 dimensions each and ReLU activations.

**Grid-based U-Net.** For the grid-based U-Net we also employ the same architecture as for the query-variation but double the number of channels to account for the larger number of predictions that the network has to produce at once. The U-Net is trained to predict all 300 velocity fields at once.

## C.3 Baseline Methods

**RN18 + Fourier Neural Operator (FNO).** A 1d FNO as constructed by [15] takes as input $\mathbf{x}$ processed by a linear layer to size 300, the number of frequencies to be predicted. We keep 32 modes and use eight FNO blocks with 128 hidden channels.

**DeepONet.** We further test a DeepONet with the RN18 as branch network and as trunk network, a four layer MLP of width 128 and 512 as output width to match the size of $\mathbf{x}$. The trunk network processes the frequency queries and is then combined with $\mathbf{x}$ to produce the prediction [11, 66]. Note, the RN18 branch network is the same as the encoder of the FQO-RN18.

**FNO (velocity fields).** The FNO takes as input the geometry interpolated to the resolution $40 \times 60$. The 2d FNO then consists of eight FNO blocks with 128 hidden channels and finally 300 output channels to map to the 300 velocity fields in this resolution. In the FNO blocks, 32 modes are preserved after the Fourier transform. Scalar parameters are introduced by a film layer after the first FNO block.

## C.4 k-Nearest Neighbors ($k$-NN)

We further test a $k$-Nearest Neighbors algorithm as a baseline, which predicts the frequency response of a plate as the mean frequency response of the $k$ closest plates in the training set. To determine the distance between different plate designs, we use the cosine distance on the 96-dimensional latent space of a convolutional autoencoder [73] trained on the beading pattern geometries. The normalized scalar properties are appended to the latent space to include them. To obtain a prediction, the frequency responses of the $k$ neighbors are averaged and the optimal $k$ in the range $[1, 25]$ is empirically determined to minimize the MSE. Determining the nearest neighbors directly in the geometry space was tried out, but yielded worse results.

## C.5 Training

The networks are trained using the AdamW optimizer [74], with $\beta = [0.9, 0.99]$ and weight decay set to 0.00005. We further choose a cosine learning rate schedule with a warm-up period [75] of 50 epochs. The maximum learning rate is set to 0.001, except for the UNet and ViT architectures, for which it is set to 0.0005. In total, the networks are trained for 500 epochs. As a validation set, 500 samples from the training dataset are set and excluded from the training and the checkpoint with the lowest MSE on these samples is selected. We report evaluation results on the previously unseen test set.

**Compute resources.** All trainings reported in this work were computed on a cluster with single A100 GPUs. The most resource intense training run took roughly 1d and 16h on a single A100 GPU and was the ablation of the number of channels with the highest scaling factor for the FQO-UNet method detailed in Section D. All other training runs with the FQO-UNet were completed in less than 24h. The trainings for the other methods were substantially shorter with i.e. the Grid-UNet finishing training in roughly 2h - 3h and likewise the FNO (velocity field) method. The roughly 20 to 30 training runs for the FQO-UNet dominate the required compute resources. We estimate that it took in total 1 A100 for 30 days. In addition, preliminary and failed experiments required further computational resources.

# D   Ablations

We provide ablation results for the loss function for training methods that predict the velocity field. We consider the loss function $L_{total} = \alpha L_{\mathbf{V}} + (1 - \alpha)L_F$ and provide results in Table 8. This ablation was performed with training batches consisting of 300 frequencies per geometry, instead of

a subset. We find that the loss on the velocity field prediction $L_{\mathbf{V}}$ is more important than the loss on the frequency response.

Table 8: Ablation of $\alpha$ value for weighing loss on the predicted velocity field vs. predicted frequency response. Higher $\alpha$ value indicates more weight on velocity field. 1 indicates only velocity field loss. The selected $\alpha$ parameter is printed in bold.

| | **V-5000** | | | |
|---|---|---|---|---|
| $\alpha$ | $\mathcal{E}_{\text{MSE}}$ | $\mathcal{E}_{\text{EMD}}$ | $\mathcal{E}_{\text{PEAKS}}$ | $\mathcal{E}_{\text{F}}$ |
| 0 | 0.25 | 8.02 | 0.15 | 4.4 |
| 0.5 | 0.15 | 5.52 | 0.11 | 2.9 |
| 0.9 | 0.09 | 3.90 | 0.08 | 1.8 |
| **1** | 0.09 | 4.00 | 0.07 | 1.8 |

We provide ablation results on the number of channels in the FQO-UNet and Grid-UNet architectures in Table 9. For the FQO-UNet, this ablation was performed with training batches consisting of 300 frequencies per geometry, instead of a subset. We find that increasing the number of channels did not lead to a perfomance improvement for the Grid-UNet and leads to marginal further improvements for the FQO-UNet architecture.

Table 9: Ablation of number of channels of FQO-UNet and Grid-UNet. The number of channels is multiplied by a constant factor over the depth, named width. The selected width parameter is printed in bold.

| | **V-5000** | | | |
|---|---|---|---|---|
| width | $\mathcal{E}_{\text{MSE}}$ | $\mathcal{E}_{\text{EMD}}$ | $\mathcal{E}_{\text{PEAKS}}$ | $\mathcal{E}_{\text{F}}$ |
| FQO-UNet | | | | |
| 16 | 0.12 | 5.00 | 0.09 | 2.2 |
| **32** | 0.09 | 3.90 | 0.08 | 1.8 |
| 64 | 0.08 | 3.82 | 0.07 | 1.8 |
| Grid-UNet | | | | |
| 16 | 0.31 | 11.44 | 0.31 | 3.9 |
| 32 | 0.24 | 8.56 | 0.25 | 3.3 |
| **64** | 0.19 | 7.57 | 0.24 | 2.7 |
| 128 | 0.24 | 8.17 | 0.21 | 3.7 |

# E  Additional Results

## E.1  Sample Efficiency

To provide full baseline results for the training with a reduced amount of samples, we refer to Table 10.

Table 10: Test results for different training dataset sizes for both settings, V-5000 and G-5000.

| | V-5000 | | | | G-5000 | | | |
|---|---|---|---|---|---|---|---|---|
| | $\mathcal{E}_{\text{MSE}}$ | $\mathcal{E}_{\text{EMD}}$ | $\mathcal{E}_{\text{PEAKS}}$ | $\mathcal{E}_{\text{F}}$ | $\mathcal{E}_{\text{MSE}}$ | $\mathcal{E}_{\text{EMD}}$ | $\mathcal{E}_{\text{PEAKS}}$ | $\mathcal{E}_{\text{F}}$ |
| FQO-UNet | | | | | | | | |
| 10 % | 0.55 | 15.26 | 0.37 | 6.4 | 0.54 | 22.70 | 0.37 | 11.1 |
| 25 % | 0.33 | 12.00 | 0.24 | 4.2 | 0.33 | 16.67 | 0.17 | 7.3 |
| 50 % | 0.19 | 8.54 | 0.17 | 2.9 | 0.19 | 11.02 | 0.12 | 4.7 |
| 75 % | 0.13 | 6.95 | 0.13 | 2.3 | 0.14 | 9.14 | 0.10 | 3.8 |
| Full dataset | 0.08 | 4.24 | 0.07 | 1.7 | 0.11 | 7.47 | 0.08 | 3.1 |
| FQO-RN18 | | | | | | | | |
| 10 % | 0.73 | 19.44 | 0.49 | 7.9 | 0.65 | 27.57 | 0.55 | 13.9 |
| 25 % | 0.58 | 14.27 | 0.31 | 7.2 | 0.45 | 21.04 | 0.44 | 8.9 |
| 50 % | 0.45 | 12.20 | 0.20 | 6.4 | 0.35 | 17.00 | 0.15 | 7.2 |
| 75 % | 0.37 | 10.96 | 0.18 | 5.5 | 0.29 | 15.06 | 0.15 | 5.9 |
| Full dataset | 0.32 | 10.70 | 0.17 | 5.3 | 0.24 | 13.51 | 0.13 | 5.1 |

Full results on training with different ratios of frequencies and geometries are provided in Table 11.

Table 11: Data generation experiment.

| | | V-5000 | | | |
|---|---|---|---|---|---|
| # Freqs | # geometries | $\mathcal{E}_{\text{MSE}}$ | $\mathcal{E}_{\text{EMD}}$ | $\mathcal{E}_{\text{PEAKS}}$ | $\mathcal{E}_{\text{F}}$ |
| 300 | 500 | 0.48 | 13.16 | 0.32 | 5.7 |
| 150 | 1,000 | 0.31 | 11.18 | 0.22 | 4.3 |
| 30 | 5,000 | 0.12 | 8.74 | 0.14 | 2.1 |
| 10 | 15,000 | 0.10 | 11.18 | 0.17 | 1.6 |
| 3 | 50,000 | 0.10 | 11.47 | 0.20 | 1.4 |
| 15 | 50,000 | 0.02 | 4.06 | 0.04 | 0.08 |
| **original** 300 | 5,000 | 0.08 | 4.24 | 0.07 | 1.7 |

## E.2  Multiple Trainings with Random Splits

To provide a notion of the variability of the results for different training runs, we performed four trainings for the FQO-UNet with different initial seeds and different random splits in training and validation set. These trainings were performed with training batches consisting of all 300 frequencies per geometry, instead of a subset. In Table 12 we report evaluation results on the respective validation sets and on the unseen test set and observe only modest variation.

Table 12: 4 models were trained on random splits in training and validation sets (4500 and 500 samples respectively). The results are denoted as mean [standard deviation].

| | V-5000 | | | |
|---|---|---|---|---|
| Evaluation set | $\mathcal{E}_{\text{MSE}}$ | $\mathcal{E}_{\text{EMD}}$ | $\mathcal{E}_{\text{PEAKS}}$ | $\mathcal{E}_{\text{F}}$ |
| Validation set | 0.096 [0.0023] | 4.1 [0.072] | 0.081 [0.0071] | 2 [0.061] |
| Test set | 0.094 [0.0031] | 4.1 [0.091] | 0.08 [0.003] | 1.9 [0.096] |

## E.3  Visualizations

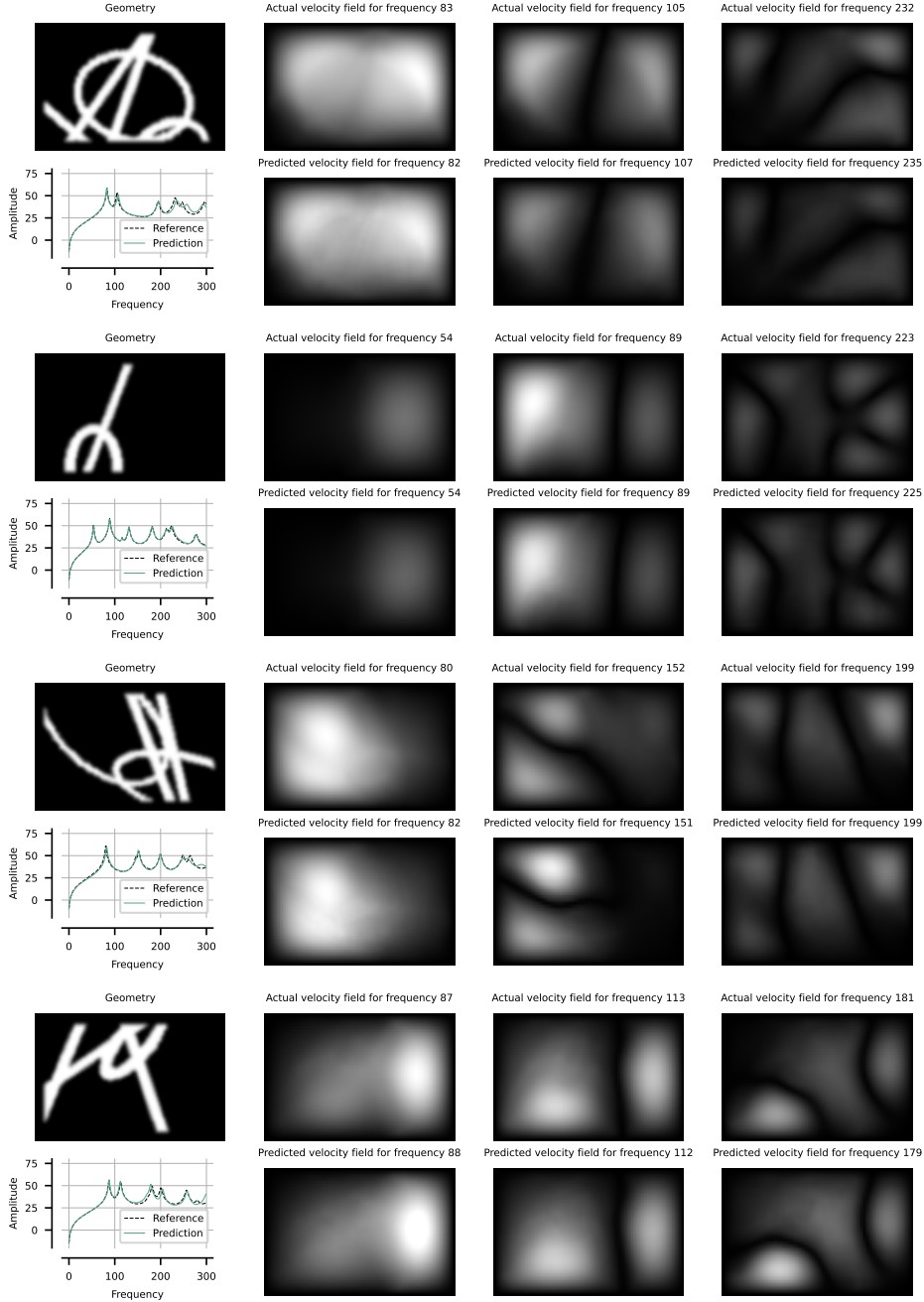

Figure 11: V-5000 example predictions. The velocity fields at the three peaks with the highest amplitude are shown. The plots are scaled with respect to the maximum velocity in the prediction and reference velocity field to make the differences in magnitude visible.

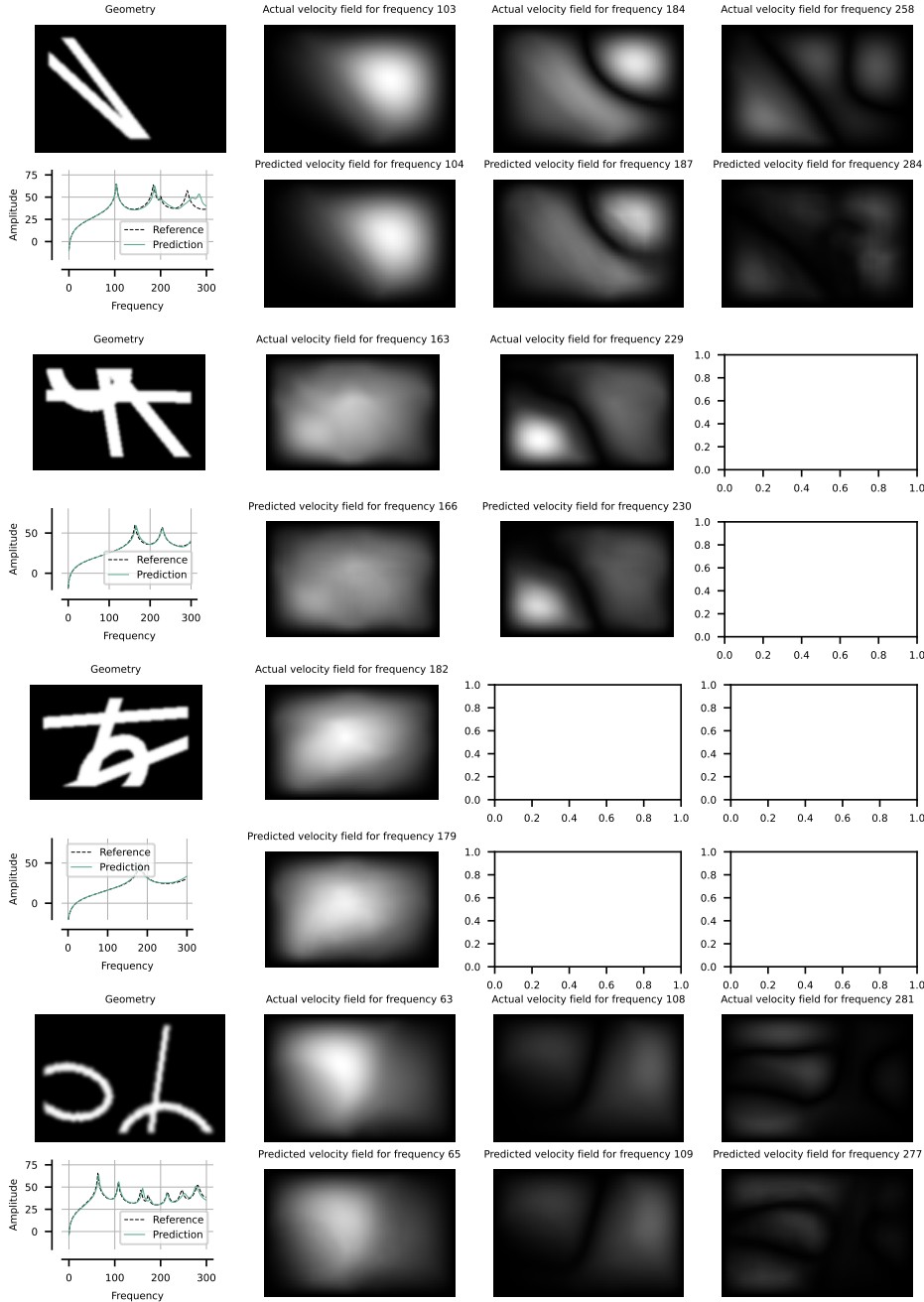

Figure 12: G-5000 example predictions. The velocity fields at the three peaks with the highest amplitude are shown. Empty axes indicates less than three peaks in the response. The plots are scaled with respect to the maximum velocity in the prediction and reference velocity field to make the differences in magnitude visible.

