# OpenReview forum: "Learning to Predict Structural Vibrations"
_NeurIPS.cc/2024/Conference — NeurIPS 2024 poster_

### Official Review · Reviewer_75jo · 2024-06-14

**Soundness:** 3
**Presentation:** 3
**Contribution:** 2
**Rating:** 5
**Confidence:** 4

**Summary:**

The paper suggests a method of using operator network theory in order to form predictions of the resultant steady state vibrational patterns which appear on a beaded flat plate under several levels of external excitation. They firstly use numerical structural engineering software (rooted in FEMs) to generate a diverse set of input-output relationships over these flat plates and the corresponding (i) Velocity fields, and (ii) The frequency response functions (FRF). The flat plate model chosen for the simulations is based on Mindlin plate theory. This forms the basis of a benchmark dataset. They then they use a bespoke deep operator network in order to derive the input-output relationships with respect to the functional operator framework.

**Strengths:**

- I think the largest strength of this paper is that of the development of the benchmark framework based upon 2D flat plate theory. It is unfortunate that many structural / mechanical / aeronautical datasets are not as publicly available, and that the problems faced in these fields are no where near as well explored when compared against those seen in CV and NLP fields, means that  the addition of an engineering benchmark 2D flat plate dataset is very welcome.

 - The fact that several other methodologies were used on the benchmark dataset, and that the (anonymously shared) code is neatly written is great.

- Additional commentary in the appendix is welcome.

**Weaknesses:**

- It appears that the proposed numerical plate simulations are still somewhat simplistic, as for example there is no high frequency noise present in FRFs (which would result from experimental limitations of sensors). Moreover there is no investigation of difficult FRF phenomenon which may arise in real life such as the appearance of closely spaced modes (which are sometimes to within 0.1 Hz) of one another. Whilst such a setting may not arise according to geometry + Midlin theory, the current wide spacing of the FRF modes makes the problem somewhat removed from the real-life difficulties that would be seen as incredibly valuable.

 - The proposed FQO-UNet does not feel like a strong original contribution. It is constructed very much from "commercial off the shelf/plug-and-play"-type NN components.

 - The necessity of the FQO-UNet model having to interpolate (predict within a neighbourhood) for computational reasons when demonstrating the FRF plots at varying frequencies doesn't seem like the most adaptable solution for real life settings. It could miss closely spaced modes, and in the presence of noise perhaps introduce spurious modes.

**Questions:**

- Are there any plans to extend the nature of the dataset? Perhaps to different plate geometries (I see you have referenced this point in your limitations section -- but do you have intentions to move in this direction)? Perhaps also to include the effect of damage in the plate? A simple model could be that of punched holes in the plate represent a highly discrete form of damage, which could then also extend the range of tasks available to your benchmark to being that of binary classification.

**Limitations:**

- Since it would feel that the main contribution is that of the benchmark dataset generation I believe a good step forward would be to generate it even comprehensively with respect to various machine learning tasks. For example, right now the notion of FRF and velocity field prediction is performed. Perhaps also the transfer learning section can be enhanced. A small study is performed with respect subset design spaces, but in structural engineering a major point would be something akin to taking a pre-trained model on bridge A, and zero/one shot applying it to bridge B -- a different structure. This may be akin to demonstrating that you can take pre-trained models on a square plate and transfer onto an entirely different geometry such as a triangular or circular plate.

 - It is understood that training took place with a lot of available data. In engineering, oftentimes in real life we do not have that much data on the real world structures. It would be good to show that some pre-traind model on the suggested computational dataset, can be extended to a real world model with an absolute minimal amount of real world data (low data cardinality)

 - Moreover in terms of pre-training, it is understood that for example even the available computational training data for fluid flow simulations within Navier Stokes (Large Eddy Simulations) is dearth. It would be good to analyze what is the absolute minimal amount of computational training data necessary in order to produce a maximal amount usability from the proposed operator training models, especially as it pertains to model transferability.

I understand these are all very difficult problems within the engineering field, but some commentary and addressing of these problems would be good.

---

> ### Author Rebuttal · Authors · 2024-08-06
>
> Dear reviewer,
>
> thanks for taking the time to write your detailed and helpful review and for recognizing the contribution of our benchmark, given that there are scarcely any publically available in the mechanics domain.
>
> **General Comments:**
>
> > The necessity of the FQO-UNet model having to interpolate [...] doesn't seem like the most adaptable solution for real life settings.
>
> We are able to drop this constraint. Please see Section 3 of our general answer for details.
>
> > It appears that the proposed numerical plate simulations are still somewhat simplistic, as for example there is no high frequency noise present in FRFs
>
> We agree that considering noisy data is an interesting task for applications like structural health monitoring, where sensor data is available. However, we target design space exploration / optimization of a numerical model. In this setting, we do not have noisy data, but deterministic simulation data in a high dimensional design space.
>
> > Moreover there is no investigation of difficult FRF phenomenon which may arise in real life such as the appearance of closely spaced modes [...]
>
> Conceptually, our model directly predicts deflection shapes, which are a superposition of all modes. Even for frequency responses with closely spaced peaks, the underlying physics and governing equations apply irrespectively.
> For modeling frequency responses with dense modes, our frequency query approach is suitable since it allows us to sample the FRF at arbitrary resolution.
>
> > The proposed FQO-UNet does not feel like a strong original contribution. It is constructed very much from "commercial off the shelf/plug-and-play"-type NN components.
>
> We kindly disagree on this point. Our FQO-UNet results from systematic experimentation and exploration specifically for vibration prediction. It involves established as well as unconventional components, but the combination of them into a working system is far from trivial. These are several crucial choices:
>
> * Transformation procedure between frequency response and velocity fields: The choice of our exact transformation (Eq. 1 in paper and code) is crucial to avoid numerical issues and enable gradient flow.
> * Balance compute between the shared embedding of the plate geometry and the frequency-specific decoder.
> * Strategy for incorporating scalar geometry parameters and frequency query: The geometry parameters are inserted in earlier layers, to affect the shared embedding for the plate geometry. FiLM layers lead to a better generalization than concatenating parameters or sinusoidal embeddings.
> * Use of self-attention: The global information exchange through self-attention support the prediction, since vibration patterns depend on the global plate geometry.
>
> Our model clearly outperforms established architectures, like the Fourier Neural Operator and DeepONet. We believe that building on established components is indeed one of the reasons for the impressive progress in (applied) machine learning in recent years.
>
> If you disagree with this perspective, we gladly elaborate on this.
>
>
> **Questions:**
>
> > Are there any plans to extend the nature of the dataset? Perhaps to different plate geometries [...]? Perhaps also to include the effect of damage in the plate? [...]
>
> We extended the G5000 setting by variable boundary conditions and force position, please see Section 1 of the general answer. In the future, we intend to move to more complicated structures, such as lightweight plate structures with additional frames and stringers. Modeling structure-fluid interactions to study sound radiation is an additional area of interest.
>
> Damage prediction would require taking the frequency response as an input for a classifier. While this is an important application, it is not directly compatible with our model design which maps from plate geometries to frequency responses.
> Thus, we mainly consider the design optimization task (general answer Section 2) which is directly applicable to our existing dataset.
>
> **Limitations:**
>
> We identified two main points in the answer: transfer learning/domain shift and sample efficiency.
>
> Transfer Learning / Domain Shift:
> > Perhaps also the transfer learning section can be enhanced. [...] taking a pre-trained model on bridge A, and zero/one shot applying it to bridge B [...] pre-traind model [...], can be extended to a real world model [...]
>
> We concur on the significance of transfer learning and conduct an additional experiment assessing transfer learning between from a more constrained dataset to a more general dataset (see Section 1 general answer).
> We hope you understand that addressing transfer to entirely new geometries and real world data is not possible within the scope of this paper.
>
> Sample Efficiency:
> > [...] It would be good to analyze what is the absolute minimal amount of computational training data [...]
>
> We agree that data-efficiency is crucial for engineering problems and generate a new dataset consisting of 50,000 plate geometries (V5000 setting) but only 15 frequency evaluations per geometry. We train models on subsets with a fixed amount of 150,000 data points by varying the ratio of frequencies and plate geometries (Table 4, Figure 2 PDF). Our original dataset has 1.5 Mio data points.
>
> Reducing the frequencies per geometry drastically increases the data efficiency of our method. With a tenth of data points, the MSE metric approaches the value of the original dataset, with slightly less favorable results for the other metrics. When training with the whole new dataset, half the size of V5000, the MSE is less than a third of our original model. Note, this strategy is only compatible with a frequency-query approach.
>
> **Table 4**
> |# freqs.|# geometries|MSE|EMD|E_Peaks|E_F|
> |---|---|---|---|---|---|
> |300|500|0.48|13.16|0.32|5.7|
> |150|1,000|0.31|11.18|0.22|4.3|
> |30|5,000|0.12|8.74|0.14|2.1|
> |10|15,000|0.10|11.18|0.17|1.6|
> |3|50,000|0.10|11.47|0.20|1.4|
> |15|50,000|0.02|3.61|0.04|0.08|
> |original||0.08|4.24|0.07|1.7|

---

> > ### Comment · Reviewer_75jo · 2024-08-13
> >
> > I believe the majority of my queries have been addressed adequately in the reply. I think the detail you mentioned of
> >
> > > Transformation procedure between frequency response and velocity fields: The choice of our exact transformation (Eq. 1 in paper and code) is crucial to avoid numerical issues and enable gradient flow.
> >
> > was somewhat lost on me upon reading, and it would be good if more detail and significance is talked about in relation to Eq 1, especially in the enabling of gradient flow.
> >
> > I agree that a lot of what I am asking for (or have general queries about) tend to be more pushing towards the boundaries of realistic engineering settings, and that this current paper tries its best to remain grounded and fundamental in its approach, so that it may be used as a spring board for future studies. And that therefore some of the ideas "I floated" may be too much too soon, however I appreciate the effort the authors went through to try to address these as such and to provide details of a few additional experiments.
> >
> > Based on this I am willing to move my score from 4 --> 5, however I am hoping that the details of this new experiment are placed in the camera ready version of the paper, that the significance of Equation 1 is expounded upon a little more, and the "fundamental-ness" of the paper (i.e. the addressing of queries such as my own including noise / SHM applications / general real life issues and how they don't readily pertain to this particular study) are addressed or discussed about in some minor capacity so that readers better understand the overall angle of this paper.

---

> > > ### Author Response · Authors · 2024-08-14
> > > **Thank you for your answer!**
> > >
> > > Thank you for your answer and raising the score. We appreciate your perspective of our paper as a 'spring board' for deeper investigations of more advanced settings and plan to investigate such challenges in the future.
> > > The additional experiments and discussions will be included along with further explanation of the transformation (Eq. 1 + code) in the camera-ready version. One aspect is that the log space enables the loss (and thus gradients) to be sensitive in off-peak regions of the frequency response.

---

### Official Review · Reviewer_YeHT · 2024-07-08

**Soundness:** 3
**Presentation:** 2
**Contribution:** 2
**Rating:** 5
**Confidence:** 5

**Summary:**

This work presents a benchmark dataset of 12,000 rectangular plate geometries with different beading patterns and their corresponding vibrational responses. The authors suggest that the dataset can be used to construct surrogate models to aid in the design and optimization of plate structures for noise reduction purposes. In addition to the dataset, the authors propose evaluation metrics to measure prediction accuracy: mean squared error, earth mover's distance, and peak frequency error. Lastly, they introduce a new neural network architecture that can map plate geometry and excitation frequencies to vibration patterns.

**Strengths:**

The paper introduces a novel dataset and neural network architecture.

**Weaknesses:**

The study is limited in scope regarding the forcing terms, material properties, and boundary conditions. It focuses exclusively on rectangular plates with elliptical and linear beading patterns.
It is unclear how the beading patterns are integrated into the Mindlin differential equation.
Extending the methodology to more complex systems is not straightforward, and the assumption of simply supported boundary conditions is restrictive.
The practical significance of predicting vibration patterns is not well-justified. In many applications, the frequency response of the plate is more critical than the detailed vibration modes.
The motivation for the study is weak and lacks clear justification.

**Questions:**

Can the authors please justify their comparisons to the baselines?
Why are these specific architectures chosen?
How do they differ?
Why do these networks have gaps in performance?

**Limitations:**

Although the authors outline many limitations of their approach, they do not address the challenge of extending their neural network architecture to different geometries. The current design relies on an image-like grid, which is not easily adaptable to other shapes or forms.

---

> ### Author Rebuttal · Authors · 2024-08-06
>
> Dear reviewer,
>
> thank you for taking the time to write your thoughtful review. We appreciate that you recognize our dataset and neural network architecture as novel, as the area of engineering and more specifically vibration prediction is not well established in the ML community. Please also take a look at our general answer, where we detail some additional results and discuss extensions to our dataset.
>
>
> **General Comments:**
>
> >The study is limited in scope regarding the forcing terms, material properties, and boundary conditions.
>
> Please note that the G5000 setting already consists of variable material properties (see Tab. 4 in the appendix). However, we agree with your assessment that a flexible model is desirable and extended the G5000 setting by variable boundary conditions and force position. At this complexity level of our dataset, there are already several challenging methodological questions in e.g. data efficiency, design optimization and transfer learning specifically for frequency response data. We believe developing an understanding of these issues is easier with our extended dataset than a more complex dataset (please see Section 1 general answer for further discussion).
>
> **Possible misunderstandings and lack of clarity:**
>
> > It is unclear how the beading patterns are integrated into the Mindlin differential equation.
>
> Technically, beading patterns define the mid-surface of the global plate geometry. This geometry is discretized via the FEM method using shell elements. Shell theory combines a plate formulation (Mindlin differential equation) and a disk formulation for in-plane loads. Thus, the beading patterns define the global position and orientation of the shell elements. Hence, the beading patterns do not have to be incorporated explicitly in the Mindlin differential equation, since the equation models the local behavior of a Shell element.
>
> >The practical significance of predicting vibration patterns is not well-justified. In many applications, the frequency response of the plate is more critical than the detailed vibration modes.
>
> We agree that from a practical standpoint, the frequency response function (FRF) is an important quantity. Because of this, we evaluate our method only on FRFs (metrics). However, for training, a clear result from our experiments is that FRF predictions become more accurate when predicting velocity fields and then directly calculating the FRF from the fields. We hypothesize that velocity field prediction acts as a regularizer and prevents (some) physically impossible solutions.
> Furthermore, from an acoustics perspective, the vibration pattern is crucial to predict the sound radiation characteristics. The normal velocity field of a vibrating structure induces pressure waves in the surrounding fluid. Assuming a weak coupling of the fluid-structure interaction, our DL model could be used to model the Neumann boundary condition of an adjacent fluid domain. This is only possible if we predict the velocity field.
>
> > The motivation for the study is weak and lacks clear justification.
>
> As acknowledged by other reviewers (1y6b, 75jo), vibration prediction is an important problem. Specifically, panel structures, which can be simulated using our approach, are used in mobility, civil engineering, renewable energy technologies, household appliances and many more. By providing a method for faster vibration mode prediction, we can contribute to the noise-reduced designs. Our benchmark dataset and method represents a first step in applying deep learning based surrogate models to these vibration prediction problems.
>
> Thank you for drawing our attention to these points. This will enable us to improve the relevant sections of our paper.
>
>
> **Questions:**
>
> >Can the authors please justify their comparisons to the baselines? Why are these specific architectures chosen? How do they differ? Why do these networks have gaps in performance?
>
> There have been some prominent and successful works in the machine learning for differential equations community, where we position our work in a broader sense. Fourier Neural Operator and DeepONet are arguably among the most well-known methods in this space, which is why we choose these methods as baselines. The k-Nearest Neighbor approach serves as a tool to gauge the complexity of the learning task: we expect a deep learning-based method to perform significantly better than k-nearest neighbors.
> The remaining architectures were constructed to investigate central architecture components based on our analysis of the problem (our paper, Section 3 Q1 to Q3). The reasoning why we assumed e.g. that the frequency-query approach is better than predicting the response for all frequencies at once is that there is intense variation between consecutive frequencies and with frequency queries the neural network can focus on single frequencies for one forward pass. In contrast to the baseline methods, our specific architectures are designed for the problem at hand, and have useful inductive biases such as the frequency query approach.
> Feel free to ask if you have further questions regarding our choice of baselines.

---

> > ### Comment · Reviewer_YeHT · 2024-08-13
> >
> > Thank you for the thorough response.
> > I have revised my score.

---

### Official Review · Reviewer_ifbc · 2024-07-13

**Soundness:** 3
**Presentation:** 3
**Contribution:** 3
**Rating:** 7
**Confidence:** 3

**Summary:**

This paper proposes a new surrogate deep learning model, the Frequency-Query Operator (FQO), designed to study structural vibrations in excited plates by mapping these plates as well as specific excitation frequencies to the resulting vibrations patterns. It introduces a new benchmark featuring 12000 plate geometries with varying geometric and material properties, and associated velocity field responses to excitations computed numerically using the finite element method. The FQO’s performance is then compared with numerous other architectures.

**Strengths:**

-	The tackling of a new time-independent problem in solid mechanics using deep learning
-	The introduction of a new FQO architecture to infer the structure vibrations, showing better performance than other classic neural operator models and a high speedup when compared against classical FEM.
-	A new 12,000 element benchmark data-set of plate responses to various frequency excitations, for different plate geometries and material properties
-	The paper is well written and organized

**Weaknesses:**

-	Only very simple plate geometries (rectangular shaped) are considered here.

**Questions:**

-	How does the current architecture scale to larger grids (finer FEMs), which would be used for instance to capture/represent smaller beadings, both in terms of accuracy and runtime ?
-	Would adding a physics loss based on equation be feasible and yield better accuracy ?

**Limitations:**

-	New encoder/decoders would need to be designed to deal with more complex geometries
-	Costly FEM simulations would need to be run on complex geometries to extend the dataset in this case

---

> ### Author Rebuttal · Authors · 2024-08-06
>
> Dear reviewer,
>
> thanks for taking the time to write your informative and helpful review. Your recognition of the contributions of our work as well as finding it well written and organized is highly appreciated!
>
> Please also take a look at our general answer, where we detail some additional results and discuss extensions to our dataset. There, we also discuss our reasoning for keeping our plate geometries comparatively simple. As you say, to extend the dataset to more complex geometries, costly simulations would need to be run. Our research currently focuses on data-efficiency, including active learning methods to keep the simulation costs reasonable, while extending our approach to more complex geometries.
>
>
> **Questions:**
>
> >How does the current architecture scale to larger grids (finer FEMs), which would be used for instance to capture/represent smaller beadings, both in terms of accuracy and runtime ?
>
> Predicting the velocity fields at a higher resolution requires adding additional upscaling layers, which would cause moderate increases in runtime. The speed-up of one neural network prediction compared to one FEM simulation will increase, since the computational cost of the FEM increases drastically with increased number of degrees of freedom. In terms of accuracy, we would not expect large changes in prediction quality. As the velocity fields are spatially smooth, no additional information would be added. In the investigated frequency range it is expected that very small beadings also only have very small effects on the vibration patterns and therefore not affect the accuracy of prediction much.
>
> >Would adding a physics loss based on the equation be feasible and yield better accuracy ?
>
> Adding a physics based loss and setting up a physics informed neural network (PINN) is a compelling idea and might yield better accuracy. However, using a PINN for shell structures is not straightforward, since it requires solving the shell equations on a non-Euclidean domain. Current research has investigated such PINNs for solving a single Shell model and compares it to classical FEM. However, extending the approach to varying geometries, like in our case the different beading patterns, remains an open challenge.
>
> We will add this discussion to the paper.

---

> > ### Comment · Reviewer_ifbc · 2024-08-12
> > **My questions have been addressed**
> >
> > Thank you for these clarifications and for the rebuttal. I maintain my score as is for the review.

---

### Official Review · Reviewer_1y6b · 2024-07-15

**Soundness:** 3
**Presentation:** 3
**Contribution:** 3
**Rating:** 7
**Confidence:** 5

**Summary:**

The paper report development of a surrogate model for prediction of the structural vibrations.  The paper reports outperformance of the their method to physics informed architectures such as DeepOnet.

**Strengths:**

The authors tackles and interesting and important problem in engineering domain which is prediction of structural vibration. This problem traditionally was solved by means of numerical techniques which is computational costly. A surrogate model can enhance the time to solution.

**Weaknesses:**

It is not clear well if such surrogate model is robust to domain shift. This means what happen if we expose to model to the structural vibrations at different scale (lower or higher) comparing to what has been used in the training phase.

**Questions:**

Does the network learn the vibrations physics or just learn the mapping from the input data to the output? Does it learn to solve a second order vibration ODE or it is imitating the input training data?
Can the authors explain how the surrogate model can predict the structural vibration of structures that has not been seen in the training phase?
How the can one use the model for scaling predicting for higher scale geometries? Have the authors utilized the model for extrapolations?
How would the model behave in presence of domain shift of the input data?
After the model is built, Can the model be used for frequency response prediction of a different system such as rotor dynamics systems or a ball bearing system?

**Limitations:**

I still doubt about the generalization and performance of of the model beyond the domain of the training data set. Can authors perform experiments to evaluate the performance of the model on a different system other than plates? such a rotary machine data or any other vibration data authors can identify.

---

> ### Author Rebuttal · Authors · 2024-08-06
>
> Dear reviewer,
>
> thank you for taking the time to write your thoughtful review. We are pleased that you also consider structural vibrations to be an interesting and important problem in the engineering domain. We discuss generalization and domain shift in more detail in Section 2 of our general answer.
>
>
> **Questions:**
>
> > Does the network learn the vibrations physics or just learn the mapping from the input data to the output? Does it learn to solve a second order vibration ODE or it is imitating the input training data?
>
> Our numerical model can be considered a mapping from the input space to resulting vibrations, which is approximated by our network.
> Our results in generalization, e.g. transfer learning and finding new beading patterns designs with properties outside the training data distribution (see General Answer Section 2), suggest that some physical knowledge is acquired.
>
> > Can the authors explain how the surrogate model can predict the structural vibration of structures that has not been seen in the training phase?
>
> Conceptually, analogously to other application fields, with the right inductive biases (e.g. predicting velocity fields, simplicity bias of SGD) a neural network learns a function that
> approximates the ground truth function (in our case numerical simulation).
> Empirically, we find our model to generalize to unseen beading patterns within distribution (Table 1 general answer) and out-of-distribution (Table 2).
>
> > How the can one use the model for scaling predicting for higher scale geometries?
> > Have the authors utilized the model for extrapolations? How would the model behave in presence of domain shift of the input data?
>
> We agree, that these are interesting and challenging questions. In Section 4 of the paper, we split our dataset (based on the number of mesh elements that are part of a beading or not) and trained either on the 'more beadings' or 'less beadings' subset while evaluating on the other set. Although the performance decreased, our networks were still able to perform better than several baseline methods despite never having seen data from this distribution.
> Further, we report results from taking a pre-trained model on V5000 and fine-tune it on G5000 (Table 2 in Section 2 general answer). As G5000 includes a substantially larger design space than V5000, the gain in performance is promising for the potential of fine-tuning a model under domain-shift.
> Without fine-tuning, we believe large extrapolations will fail with the current setup.
>
> >After the model is built, Can the model be used for frequency response prediction of a different system such as rotor dynamics systems or a ball bearing system?
>
> To extend our model to applying to systems like ball bearings and rotors, we would have to construct it differently. Our network depends on the specific input space defined by the beading pattern and scalar parameters of a plate. A true foundational model capable of simultaneously modeling plates, rotors and ball bearings would need to map all these systems to a common input space. Unfortunately, we are not yet at this point. However, our method could be used to simulate plate components in such systems, where rotating components can be modeled as harmonic excitations.

---

> > ### Comment · Reviewer_1y6b · 2024-08-08
> > **My questions are answered and addressed. Additional comments added.**
> >
> > Please make sure in the revised version, you include a section to address the response to the questions asked which would be very beneficial for the future readers to have an understanding on the limitations of the approach (Domain shift evaluation, applicability to other datasets, scalability limitations and etc.)
> > Please, if possible, identify a methods to quantify the difference in the distribution of the training data set and the tested unseen data set, so that it support your statement on applicability of the NN to an unseen enough different data set. Metrics for quantification of the distance between two distribution such as relative entropy might be useful for you.
> > Please also describe completely the architecture of the tested DeepOnet model you tested and the details of the training and evaluation of the physics informed network. This is of outmost importance for reproducibility of the research be others and ease of verifications of contributions of the model.
> > A section for mentioned future works can also open new interesting topics for the community.
> > Thanks for the rebuttal. I keep my score as is for this review.

---

> > > ### Author Response · Authors · 2024-08-09
> > > **Thank you for your answer!**
> > >
> > > Thank you for your answer. We will make sure to include any missing information in our revised version. We trained DeepONet in a data-driven manner, as described in Lu et al. (2019, 2021), where it was introduced. This will be further described in our revised version. Thank you as well for pointing out the possibility of quantifying distribution differences between different datasets, we will investigate this.

---

### Author Rebuttal · Authors · 2024-08-06

Dear reviewers,

Thank you for many valuable and thoughtful comments. We are pleased that the reviewers recognize the value of our novel benchmark (**ifbc, 1y6b, YeHT**)  and method (**YeHT, ifbc**) for the important problem (**1y6b, ifbc**) of structural vibration prediction. as well as rating the paper (**ifbc**) and code (**75jo**) well-written.

First, we would like to comment on three key points, mentioned by multiple reviewers:


## 1. Dataset complexity: Extension to more complex geometries, boundary conditions, or forcing terms (**ifbc, YeHT, 75jo**)

More complex geometries are an exciting research avenue. We deliberately constrained ourselves to rectangular plates to focus on methodological issues, as an investigation of data scarcity, design optimization and transfer learning (also suggested by the reviewers) is already challenging in this setting. This enables us to benefit from the comparatively fast numerical simulations for rectangular plates and obtain insights which we expect to generalize to more complex and less constrained geometries.

Therefore, in the scope of this paper, we opted for retaining the current plate geometries which have the direct engineering application of metal beading but extend the G5000 dataset (G5000 new) by two new aspects: (a) variable rotational stiffness at the boundary, allowing us to predict simply supported as well as clamped boundary conditions, and (b) a variable point force position. This extension makes the prediction task more challenging:

**Table 1**
| |MSE |EMD |E_Peaks|E_F|
|----|----|----|-------|---|
|G5000 - original|0.09|4.94|0.07|2.5|
|G5000 - new |0.11|7.47|0.08|3.1|


## 2. Deeper investigation of generalization, data efficiency  and transfer learning (**1y6b, 75jo**)

Generalization and robustness to domain shift are critical for surrogate models to be applicable when the mechanical model changes. We reported results on transfer between different subsets of our dataset in Table 2 in the paper. To further investigate generalization we show new results when using our FQO-UNet for (1) design optimization in conjunction with a guided diffusion method (see Figure 1 in PDF), (2) an experiment on minimizing the amount of training data for a given prediction quality, and (3) transfer learning between V5000 and G5000:
1. For design optimization, the goal is to find beading patterns with the lowest possible mean frequency response in predefined frequency ranges. Based on the gradient information from the surrogate model, our guided diffusion method is able to generate beading patterns with a mean frequency response well below any plate in the training data (verified by numerical simulation). This highlights the potential for generalization beyond the training data distribution, as the resulting beading patterns look different from the training data, with more variety in thickness, form and number of beadings.
2. We generate a new dataset consisting of 50,000 plate geometries but with only 15 frequency evaluations per geometry. Reducing the frequencies per geometry drastically increases the data efficiency of our method. With a tenth of data points compared to our original dataset, the MSE metric approaches the original value (Figure 2 in PDF, and answer to reviewer **75jo**).
3. For the transfer learning experiment, we took a model trained on V5000 as an initialization for the G5000 setting. When initializing with the pretrained model, the performance improves in all metrics (Table 2).

**Table 2**
|           |                   |MSE  |EMD  |E_Peaks    |E_F|
|----|-----|-----|-----|-----------|---|
|G5000 | from scratch |0.086|4.94 |0.068      |2.5|
|G5000 |V5000 fine-tuned|0.061|4.01 |0.053      |1.9|
|G5000 - new | from scratch |0.111|7.47 |0.079      |3.1|
|G5000 - new | V5000 fine-tuned|0.095|4.63 |0.078      |1.9|


## 3. Limitations of FQO-UNet: Grids and Frequency (**YeHT, 75jo**)

Indeed, our architecture requires a regular grid structure with respect to the input plate, constraining the input geometries (see a discussion of this setting in general answer 1).
However, this limited setting enables us to focus on understanding which aspects are relevant for frequency response prediction. For example, some of our findings are:

* The frequency query approach leads to better predictions. They also enable evaluating the FRF at arbitrary locations (e.g. in 0.1 Hz steps or beyond the training frequency range).
* Predicting a field quantity (in our case velocity) first and then directly calculating a frequency response from it is more data efficient than directly predicting the frequency response.
* Convolutions pose a beneficial inductive bias for predicting the field quantity.

These insights can now inform the development of more flexible neural network architectures.
As mentioned by **75jo**, we originally mapped to five velocity fields per query. This constraint is technically not necessary and was only introduced to speed up training.
This goal can also be achieved by training on a subset of frequencies per geometry per batch, without frequency bundling. The results (Table 3) are close to identical, while frequency subset training is around 2 times faster.

**Table 3**
|                              |MSE |EMD |E_Peaks|E_F|
|-----|-----|-----|-----------|---|
|V5000 - original|0.09|3.90|0.08   |1.8|
|V5000 - one prediction per query      |0.08|4.24|0.07   |1.7|

# Conclusion
Concluding, we consider this work a solid foundation for the vibroacoustics and ML community to make progress in applying ML methods. Interdisciplinary work like ours first needs to establish common problems and a common language, which we are grateful to the reviewers to acknowledge. With the design optimization task and the exploration of data-efficiency we display two promising future research directions of our work.

---

### Decision · Program_Chairs · 2024-09-25

**Decision:**

Accept (poster)

**Comment:**

The reviewers all recognized the value of proposed the application of surrogate models to prediction of structural vibrations, and the rich experimental setting. While they found the geometries a little simplistic, they appreciated the new proposed dataset and quality of the code released. They found the manuscript well written, and the authors addressed the reviewers questions in their rebuttal.